# Electrical Activity Changes and Neurovascular Unit Markers in the Brains of Patients after Cardiac Surgery: Effects of Multi-Task Cognitive Training

**DOI:** 10.3390/biomedicines12040756

**Published:** 2024-03-28

**Authors:** Irina Tarasova, Irina Kukhareva, Darya Kupriyanova, Tatjana Temnikova, Evgenia Gorbatovskaya, Olga Trubnikova

**Affiliations:** Research Institute for Complex Issues of Cardiovascular Diseases, Academician Barbarash Blvd., 6, 650002 Kemerovo, Russia; kuchin@kemcardio.ru (I.K.); kuprds@kemcardio.ru (D.K.); t.13.ermakova@mail.ru (T.T.); eugenia.tarasowa@yandex.ru (E.G.); truboa@kemcardio.ru (O.T.)

**Keywords:** cognitive training, brain electrical activity, neurovascular unit, S100β, BDNF, coronary surgery

## Abstract

Background: There is growing interest in finding methods to enhance cognitive function and comprehend the neurophysiological mechanisms that underlie these improvements. It is assumed that non-pharmacological interventions have better results in cognitive recovery. The aim of this study was to assess the effect of multi-task cognitive training (MTT) on electroencephalographic (EEG) changes and markers of the neurovascular unit in patients undergoing coronary artery bypass grafting (CABG). Methods: This prospective cohort study involved 62 CABG patients aged 45–75 years, 30 of whom underwent a 5–7-day MTT course. The groups of patients were comparable with respect to baseline clinical and anamnestic characteristics. An EEG study was performed before surgery and 11–12 days after CABG. Markers of the neurovascular unit (S100β, NSE, and BDNF) were examined at three time points: before surgery, within the first 24 h after surgery, and 11–12 days after CABG. Results: Patients without training demonstrated higher relative theta power changes compared to the MTT patients. The course of MTT was associated with low plasma S100β concentration but high BDNF levels at the end of the training course. Conclusions: The theta activity changes and the markers of the neurovascular unit (S100β, BDNF) indicated that the severity of brain damage in cardiac surgery patients after a short course of MTT was slightly reduced. Electrical brain activity indicators and vascular markers can be informative for monitoring the process of cognitive rehabilitation in cardiac surgery patients.

## 1. Introduction

The development and availability of advanced medical technologies have increased life expectancy. However, age-dependent diseases, including cardiovascular pathology, significantly increase the risk of cognitive impairment and disability [1,2,3]. Previous studies have shown that successful cognitive functioning depends on adequate cerebral blood flow [4,5]. Cardiovascular diseases independently lead to cerebral blood flow disorders and cognitive impairments, and may require cardiac surgery with artificial blood circulation, which can also cause acute ischemic brain damage [6,7].

The search for methods to enhance cognitive functioning and understand the neurophysiological mechanisms that underlie these improvements is growing. Non-pharmacological interventions are recognized by many experts as the most effective [8,9,10]. Recent reports indicate that combining physical and cognitive exercises enhances cognitive functions more effectively than either approach alone [11]. The multi-tasking approach, which involves performing motor and cognitive tasks simultaneously, triggers the activation of the different cognitive domains, such as attention, inhibitory control, and executive function [10,12]. Preliminary data showed that multi-task training (MTT) can provide effective cognitive recovery in cardiac surgery patients [10,13].

There is limited research on the impact of multi-tasking on different aspects of cognitive recovery. This is especially true for the recovery of cognitive functions in the postoperative period of cardiac surgery.

Digital electroencephalography (EEG) is widely used for the non-invasive control of brain activity and studying the fundamental mechanisms of brain functioning [14,15,16]. Unlike MRI and lumbar puncture, EEG is a non-invasive and inexpensive approach, commonly used in describing cognitive disorders of various origins, as it can study high brain functions with maximum time resolution [17,18]. The relevance of theta rhythms of resting EEG for differentiating patients with and without vascular cognitive disorders has been previously identified. High theta rhythm power, mainly in the posterior brain areas, has been shown to be associated with cognitive impairment in cardiac patients [19]. In the study by Deiber et al., the frontal-induced theta response was significantly reduced at baseline in progressive MCI compared to both stable MCI and controls, suggesting altered top-down attentional control [20]. Thus, theta activity changes can be viewed as a marker for impaired attention and executive functions.

Understanding neurophysiological changes associated with rehabilitation remains a burgeoning area of research. Gangemi et al. investigated the neurophysiological effects of cognitive stimulation training in the chronic phase of patients with ischemic stroke. After training, there was a significant increase in both the alpha band power in the occipital areas and the beta band power in the frontal areas, while the theta band power did not show any significant variation [21]. The effects of multi-task executive function training on the features of specific EEG microstates were examined in a study by Santarnecchi and colleagues. The left parietal and occipital regions showed an increase in current density EEG activity after cognitive training, as well as a decrease in density in the right inferior frontal gyrus and insula [22]. The authors concluded that the changes in EEG dynamics associated with successful training suggest the importance of low-level visual computational processes, as well as prefrontal control over parietal dynamics.

Peripheral blood biomarkers for brain damage can provide information about various pathological states [23,24,25]. The neurovascular unit has already been identified by several indicators, and markers that can measure its functioning are currently being examined. Among these indicators is S-100 calcium-binding protein B (S100β). This protein is mainly expressed in astrocytes, which allows it to be seen as an acute ischemia marker associated with the severity of brain damage [23,26]. In a study by Onatsu et al., it was shown that serum S100β levels are correlated with neurological deficits, infarct size, and significant brain edema during acute stroke [24].

Neuron-specific enolase (NSE) is a glycolytic enzyme that is produced in the cytoplasm of neurons and is precise enough in its ability to detect neuronal damage [27]. Neuron-specific enolase is increased in acute ischemic brain damage [28]. NSE is used as a marker for postoperative cerebral dysfunction in patients who have undergone coronary and carotid artery revascularization, but it is unclear whether it predicts cerebrovascular outcomes [29].

Also, in addition to neurophysiological methods of studying the recovery process in cognitive structures and understanding the mechanisms involved, changes in neurotrophic factors, in particular, brain-derived neurotrophic factor (BDNF), are now of interest to researchers. BDNF is believed to play a key role in brain development and maintenance [30]. Compared to other members of the growth neurotrophic factor family, BDNF is highly expressed in the cerebral cortex and hippocampus [31]. It has been shown that BDNF promotes neuroplasticity and facilitates synaptic transmission, dendritic modification, receptor trade, and long-term potentiation [30]. Moreover, BDNF supports neurogenesis, synaptic growth, and recovery [32]. In longitudinal observational studies of older persons, higher baseline BDNF levels were associated with reduced probability of dementia development [30,33]. It should be noted that BDNF changes demonstrated a close relationship with changes in the brain under pharmacological and non-pharmacological interventions, including physical and cognitive training. This is one reason why BDNF may be considered one of the most promising biomarkers for the study of the cognitive recovery processes, especially when using multi-task training.

A study by Huang et al. investigated the effects of motor control training on upper limb motor function and neuroplasticity in stroke patients. Significant improvements in overall upper limb function were found after the intervention. BDNF expression levels significantly increased only in the intervention group. The authors proposed that BDNF induction might be affected by a specific type of intervention or rehabilitation program [34].

The advantages of electrophysiological methods and neurotrophic factors suggest that they can be used to identify the key neurophysiological characteristics of successful cognitive recovery in cardiac surgery patients. The current study aimed to investigate the relationship between intervention-associated changes in markers of the neurovascular unit and EEG activity in cardiac surgery patients with and without multi-task training.

## 2. Materials and Methods

### 2.1. Patients

This study was approved by the Ethical Committee of the Research Institute for Complex Issues of Cardiovascular Diseases (protocol No. 10 dated 12 October 2020), and was in line with the Helsinki Declaration. The study design is outlined in Figure 1. The study was conducted in a prospective, observational cohort setting. A sample of 60 subjects was chosen from a group of patients who had on-pump coronary artery bypass grafting (CABG) at the Research Institute for Complex Issues of Cardiovascular Diseases clinic. Informed written consent was obtained from all participants included in the study. The inclusion criteria consisted of patients who were aged 45 years or older, had elective CABG, were right-handed to eliminate any influence of laterality on EEG, and had a Montreal Cognitive Assessment (MoCA) score ≥ 18. Standardized physical and instrumental examinations were performed on all patients involved. The examiners were not informed about the nature of the patients’ participation in the study.

The criteria for the exclusion of patients from the study were as follows:Significant pathological changes in the brain, including leukoaraiosis and cysts, as revealed through multi-layered spiral computed tomography;Diseases of the central nervous system, including stroke, epilepsy, and neuroimmunological disorders;Functional class IV heart failure according to the New York Heart Association (FC NYHA IV) guidelines;Life-threatening arrhythmias;Malignant pathology;Chronic obstructive pulmonary disease with persistent breathing difficulty;Depressive symptoms, as identified by the Beck Depression Inventory (BDI-II) (sum scores > 8).

All participants had normal or corrected-to-normal visual acuity, and none reported a history of alcohol or drug abuse. The study did not include patients who were receiving anxiolytic therapy. To avoid any concomitant acute or chronic neurological conditions that could interfere with EEG data, a standard neurological examination was performed. A 1.5 T Exelart Atlas MRI system (Canon Medical Systems, Toshiba, Tokio, Japan) was used to assess the baseline brain state in all patients. Leukoaraiosis, cysts, gliosis, and the enlargement of cerebrospinal fluid spaces were examined. The majority of patients showed grade 1–2 leukoaraiosis, with periventricular and subarachnoid space dilation. These changes are characteristic of the different stages of chronic brain ischemia. Brain cysts were a reason for the exclusion of patients from the study.

The study sample was separated into two groups: multi-task training (MTT) I (*n* = 30) and control (*n* = 32). Table 1 presents the preoperative clinical and anamnestic characteristics of the groups. The MTT patients and controls were comparable in terms of age, baseline cognitive status, educational level, and other clinical factors. The treatment protocol and medication did not show any significant differences.

### 2.2. Neurophysiological Assessment

To exclude patients with severe cognitive impairment, cognitive screening using modified Russian versions of the Montreal Cognitive Assessment Scale (MoCA) was conducted upon admission to the hospital. Subjects with MoCA scores ≥ 18 were excluded from the study.

EEGs were recorded via a 62-channel Quik-cap using a NEUVO-64 system (Compumedics, El Paso, TX, USA) 2–3 days before surgery and 11–12 days after CABG. The EEGs were recorded in the eyes-closed condition in a dimly lit, soundproof, electrically shielded room, and recording lengths were about 5 min. A modified 10/10 system was used to locate the electrodes on the scalp. The reference and ground electrodes were placed on the tip of the nose and the center of the forehead, respectively. To monitor eye movement artifacts, bipolar eye movement electrodes were placed on the canthus and cheek bone. The amplifier bandwidths were 1.0 to 50.0 Hz, and EEGs were digitized at 1000 Hz. The Neuroscan 4.5 software program (Compumedics, El Paso, TX, USA) was used to analyze the data offline using automatic algorithms. Visual inspection of the eye movements, electromyographic signals, and other artifacts were conducted. Artifact-free EEG fragments were divided into 2 s epochs and underwent Fourier transformation. For each subject, the EEG power values were averaged within the theta range (4–6 Hz). The EEG frequency band selected for the planned analysis was determined based on previous studies and the significance of low-frequency activity in brain damage [19,35]. The data from 56 leads were symmetrically averaged into 5 electrode zones in the left and right hemispheres: frontal (Fp1/2, AF3/4, F1/2, Fp3/4, Fp5/6, F7/8), central (FC1/2, FC3/4, FC5/6, C1/2, C3/4, C5/6), parietal (CP1/2, CP3/4, CP5/6, P1/2, P3/4, P5/6, P7/8), occipital (PO3/4, PO5/6, PO7/8, O1/2) and temporal (FT7/8, T7/8, TP7/8). The midline sites (Fpz, Fz, etc.) were excluded. The indicator of relative change (Δ) was calculated using the following formula: (baseline value–postoperative value)/baseline value) × 100%. The negative values of the indicator showed an increase in the theta power after the surgery compared to baseline, while the positive values showed a decrease.

### 2.3. Laboratory Data

Measurements were made at three time points in the study period (1–2 days before CABG, within the first 24 h after surgery, and 11–12 days after CABG). After a 12 h fasting period, venipuncture was used to collect whole blood samples from each patient. The procedure for obtaining serum involved allowing whole blood samples to coagulate at room temperature for 30 min, and then, centrifuging them at room temperature for 15 min at 3000 revolutions per minute. The collected serum was stored in polypropylene tubes at −70 °C until assayed. Measurements were taken for S100β, NSE, and BDNF serum parameters. The serum concentrations of S100β and NSE were quantitatively determined by using a commercially available enzyme-linked immunosorbent assay ELISA (FUJIREBIO Diagnostics, Inc., Tokio, Japan). The serum BDNF levels were measured using an ELISA kit (R&D Systems, Minneapolis, MN, USA) on a «Uniplan» tablet reader («PICON», Moscow, Russia). The intra-assay coefficients of variation were <10% for all measurements.

### 2.4. Multi-Task Training

The training course started 3–4 days after CABG and was conducted daily for 5–7 days. A postural balance task and three cognitive subtasks, including mental arithmetic, verbal fluency, and divergent tasks, were the basis of the original multi-task training protocol. To complete the postural balance exercise, trained patients stood on a balance platform and used visual feedback to keep their center of pressure in the same position. A marker on the monitor screen indicated the subject’s pressure center. The patient’s goal was to align this marker with the target positioned in the center of the monitor.

At the same time, the patient performed cognitive tasks of various types. The task of mental arithmetic was to subtract 7 from 100 in a sequential manner. The verbal fluency task involved saying as many words as possible that begin with a particular letter within a minute. In the divergent task, the participants were asked to come up with creative ways to use ordinary objects (such as a ruler, glass, and a rope). All tasks were completed one by one, with a short period of rest and breaks from the balance platform [10].

### 2.5. Statistical Analysis

The analysis of all data was carried out using Statistica 10.0 (StatSoft, Tulsa, OK, USA). The Shapiro–Wilk test was used to test for normal distribution of the data. The majority of the clinical parameters were not distributed in a normal way and were evaluated using the Mann–Whitney test. To normalize the EEG data, they were transformed using Log10. Further analysis of the EEG data was carried out using a repeated-measures ANOVA. Levene’s test was used to verify that the variances for EEG variables were equal. In the ANOVA, statistical significance was corrected using the Greenhouse–Geisser correction. Pairwise comparisons were carried out in groups of patients by using Newman-Keuls multiple comparison tests. The parameters of S100β, NSE, and BDNF after the logarithm transformation were compared with the *t*-test. Indicators within a group were compared using a dependent-samples *t*-test, and differences between groups were evaluated using an independent-samples *t*-test.

## 3. Results

### 3.1. Theta Power Data

The repeated measures ANOVA was conducted using GROUP (two levels: MTT/control) as the between-subjects factor and TIME (two levels: before and after surgery), AREA (five levels: frontal, central, parietal, occipital, and temporal), and LATERALITY (two levels: left and right hemispheres) as within-subjects factors.

Statistically significance of the factor TIME (F_1,60_ = 80.92, *p* < 0.00001, η = 0.57) and an interaction of the factors GROUP and TIME (F_1,60_ = 11.51, *p* = 0.001, η = 0.16) were found. Both MMT patients and the controls had higher values of the theta1-rhythm power at 11–12 days after CABG in comparison to the baseline values (Figure 2a). At the same time, the control group had more significant changes, which are also validated by differences in the indicator of relative changes (Δ), *p* = 0.001 (Figure 2b).

The AREA factor and interaction of the factors AREA and LATERALITY (F_1,60_ = 80.37, *p* < 0.00001, η = 0.57 and F_1,60_ = 8.11, *p* = 0.00003, η = 0.11, respectively) were statistically significant. This fact reflected the general patterns in the topographic distribution of theta activity in this sample (see Table 2). It was found that the values of the theta1-rhythm power were significantly higher in the left temporal and occipital areas in comparison to the right ones.

### 3.2. Neurovascular Unit Data

In the control group, the S100β serum concentrations increased within the first 24 h after CABG (*p* = 0.05), and then, significantly decreased 11–12 days after cardiac surgery (*p* = 0.03) (see Figure 3). There were no significant differences in the MTT group, but the S100β changes showed the same trend. However, the S100β level was significantly lower in MMT patients than in the controls 11–12 days after cardiac surgery (*p* = 0.03). The changes in serum NSE concentrations were similar; however, the difference did not reach a statistically significant level.

The BDNF serum levels decreased within the first 24 h of surgery in both the MTT group and the controls (*p* = 0.01 and *p* = 0.02, respectively). However, 11–12 days after CABG, the MTT group showed higher BDNF levels compared to the pre-operative values (*p* = 0.05). In the control group, the BDNF serum concentration continued to decrease compared to the first 24 h after CABG (*p* = 0.03). Also, the serum concentrations of BDNF were significantly higher in the MMT group than in the control group 10–11 days after CABG (*p* = 0.03).

## 4. Discussion

The results of this study demonstrate changes in the theta power, S100β, and BDNF serum levels after a short course of MTT in the early postoperative period of on-pump CABG.

It has been previously shown that on-pump surgical interventions such as CABG may lead to ischemic brain damage [36,37,38]. A magnetic resonance imaging study detected silent brain lesions after CABG in 20.1% of patients [37]. Recent research has indicated that approximately 60–70% of cardiac surgery patients experience postoperative cognitive dysfunction during the early postoperative period [10,37,38].

Event-related theta power is thought to play a role in executive control mechanisms such as conflict detection and response suppression. There have been studies that have shown a reduced theta response in older adults with cognitive decline [39]. An analysis of event-related theta synchronization/desynchronization in cardiac surgery patients found that they exhibited less theta desynchronization in the left frontal–center region compared to patients without cognitive decline [40].

On the other hand, an increase in resting-state theta activity is probably linked to cerebral ischemia during on-pump cardiac surgery. The adverse factors associated with cardiopulmonary bypass (embolism, systemic inflammation, etc.) cause neural dysfunction, tissue atrophy, and damage to neural networks, resulting in the cortical suppression of cortical areas and domination of low-frequency brain activity [38,41]. An increase in resting EEG theta power has been linked to mild cognitive impairment and dementia as diseases progress [42]. It may possibly reflect the vulnerability or resilience of subcortical and thalamocortical loops and the ascending activating systems as additional and supplementary relevant information concerning the progression of cognitive decline.

Previous research has demonstrated that brain electrical activity patterns in patients with brain ischemia have specific characteristics. Al-Qazzaz and colleagues found that the degree of EEG complexity was significantly lower in patients with vascular cognitive impairment compared to healthy subjects [43]. Zappasodi et al. showed that patients with acute unilateral stroke who have an increase in low-frequency activity and decreased hemispheric asymmetry are more likely to have poorer functional outcomes [44].

We demonstrated that the indicator of relative theta power changes was higher in patients without training (55.6%) compared to the MTT patients (25.2%). The impact of cognitive training on the functional organization and compensatory rearrangements of electrical brain activity has been previously demonstrated [22,45]. The activation of local neural networks increased during combined training (physical activity and cognitive load), suggesting the involvement of compensatory brain resources [45]. Thus, a lower decrease in theta power, as seen for the relative change’s indicator, probably indicated that the severity of brain damage in cardiac surgery patients after MTT was slightly reduced.

The changes in neurotrophic factors in our study may provide evidence for this assumption. We found an increase in S100β concentrations within the first 24 h after CABG only in the control patients. The MTT group did not show any significant differences, but the S100 changes exhibited the same trend. Although the changes in serum NSE concentrations were similar, the difference did not reach a statistically significant level.

The results of previous studies are consistent with these data. The ability of S100 to identify structural and functional brain damage in cardiac surgery patients was demonstrated by recent studies [29,46]. According to Silva et al., S100 is more closely associated with POCD development than NSE [47].

In addition, we demonstrated that all the patients showed a decrease in BDNF serum levels, regardless of MTT intervention, during the early postoperative period of CABG. This result was confirmed by a recent study [48]. The authors showed that decreased serum levels of BDNF are associated with cognitive decline in patients undergoing CABG surgery. However, we found that the MTT group showed an increase in BDNF levels 11–12 days after CABG when compared to their preoperative values. The serum concentration of BDNF in the control group remained low.

Several studies have demonstrated that BDNF levels increase during physical exercise, including a study of cardiac surgery patients [29,49]. It was indicated that mood and cognitive functions improved in elderly patients with various levels of depression who underwent cognitive training, which was accompanied by an increase in the serum concentration of BDNF [50].

In summary, it can be concluded that on-pump cardiac surgery causes complex, largely unexplored mechanisms of brain damage. The key pathogenic processes are neuroinflammation and ischemia/reperfusion of brain structures caused by cardiopulmonary bypass. The connection between neuroinflammation and BDNF has been demonstrated [51]. On the other hand, there is evidence that lower levels of BDNF are associated with decreased cognitive function in the postoperative period [52]. Thus, a study by O.H. Miniksar et al. demonstrated associations between BDNF levels and MoCA parameters in patients with CABG [48].

It can be assumed that a short MTT course triggered the processes of neuroplasticity through BDNF expression and reduced ischemic brain damage in the early postoperative CABG period. Thus, the study of the mechanisms underlying postoperative cognitive decline in cardiac surgery patients, with the search for new biomarkers that would have a high diagnostic and predictive ability, as well as reflecting the effectiveness of cognitive recovery procedures, are clinically relevant. Further research is needed to examine the functioning of the neurovascular unit during cardiac surgery.

## 5. Limitations

It is important to mention some of the limitations of this study. One limitation is the small sample of consecutive patients. Another limitation of the study is the short course of cognitive training. It is our belief that an extended training period would have a more significant impact on the analyzed indicators. Further research should be conducted to address these issues.

## 6. Conclusions

The theta activity changes and the markers of the neurovascular unit (S100β, BDNF) indicated that the severity of brain damage in cardiac surgery patients after a short course of cognitive training was slightly reduced. Electrical brain activity indicators and vascular markers can be informative for monitoring the process of cognitive rehabilitation in cardiac surgery patients.

## Figures and Tables

**Figure 1 biomedicines-12-00756-f001:**
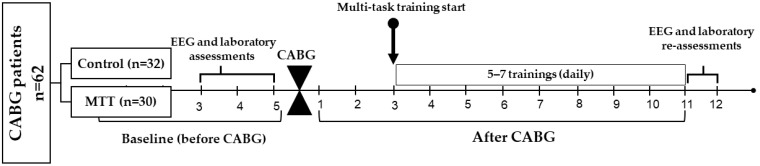
Design of the study. CABG, coronary artery bypass grafting; MTT, multi-task training.

**Figure 2 biomedicines-12-00756-f002:**
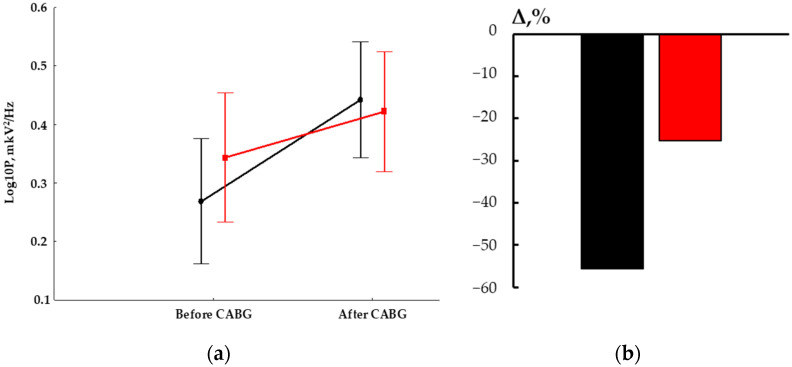
The theta rhythm power changes in patients with MMT and controls: (**a**) Log10 power values: red line—MTT group, black line—control; (**b**) the indicator of relative change (Δ): red column—MTT group, black column—controls.

**Figure 3 biomedicines-12-00756-f003:**
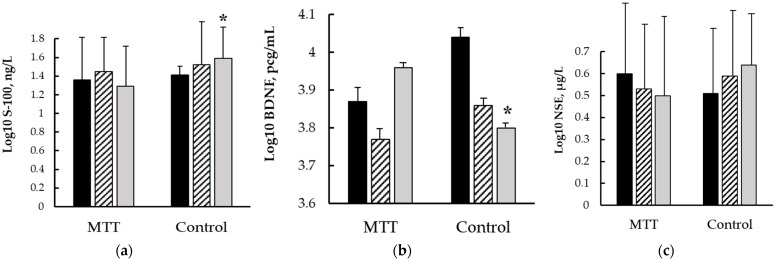
The serum concentrations of (**a**) S100β; (**b**) BDNF; and (**c**) NSE: dark columns—the pre-operative values, dashed columns—the first 24 h after surgery, light columns—the values 11–12 days after CABG, error bars denote SE, *—*p* < 0.05 in *t*-test for independent samples.

**Table 1 biomedicines-12-00756-t001:** The baseline clinical and anamnestic characteristics of the patients.

Variable	Multi-Task Training (*n* = 30)	Control (*n* = 32)	*p*-Value
Age, years, Me (Q25; Q75)	65 (60; 69)	61 (60; 65)	0.11
Male/female, *n*	6/24	6/26	0.91
MoCA, scores, Me (Q25; Q75)	26 (24; 28)	26 (23; 27)	0.61
Educational level, years, Me (Q25; Q75)	12 (11; 15)	12 (11; 14)	0.23
Functional class of angina, *n* (%)			0.63
I–II	24 (80)	24 (75)	
III	6 (20)	8 (25)	
Functional class NYHA, *n* (%)			0.86
I–II	27 (78)	30 (78)	
III	3 (10)	2 (6)	
Ejection fraction of left ventricle, %, Me (Q25; Q75)	62 (55; 68)	64 (55; 67)	0.61
Diabetes mellitus type 2, *n* (%)	9 (30)	15 (47)	0.25
Carotid artery stenoses, *n* (%)	17 (57)	10 (31)	0.07
Beta-blocker therapy, *n* (%)	30 (100)	30 (94)	0.31
Antiplatelet therapy, *n* (%)	30 (100)	32 (100)	-
ACEi therapy, *n* (%)	16 (53)	21 (66)	0.33
Statin therapy, *n* (%)	30 (100)	32 (100)	-
Diuretic therapy, *n* (%)	24 (80)	25 (78)	0.85
Cardiopulmonary bypass time, min, Me (Q25; Q75)	88 (69; 102)	72 (57; 105)	0.24
Surgery time, min, Me (Q25; Q75)	225 (175; 240)	200 (180; 245)	0.69

ACEi, angiotensin-converting enzyme inhibitor; MoCA, Montreal Cognitive Assessment Scale; NYHA, heart failure according to the New York Heart Association.

**Table 2 biomedicines-12-00756-t002:** The topographic distribution of theta activity in the CABG patients.

Brain Area	Left Hemisphere, Log10P, mkV^2^/Hz, M ± SE	Right Hemisphere, Log10P, mkV^2^/Hz, M ± SE	*p*-Value
Frontal	0.32 ± 0.038	0.33 ± 0.039	0.31
Central	0.36 ± 0.032	0.37 ± 0.032	0.1
Parietal	0.43 ± 0.030	0.43 ± 0.033	0.83
Occipital	0.47 ± 0.035	0.44 ± 0.036	0.001
Temporal	0.28 ± 0.031	0.24 ± 0.034	0.000009

## Data Availability

Data are contained within the article.

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
