# Peer review of "Electrical Activity Changes and Neurovascular Unit Markers in the Brains of Patients after Cardiac Surgery: Effects of Multi-Task Cognitive Training"

_biomedicines, 2024, doi:10.3390/biomedicines12040756_

Round 1
Reviewer 1 Report
Comments and Suggestions for Authors
[INTRODUCTION]
1. To enhance the manuscript's flow to be more consistent and natural, there is a need to diversify similar sentence structures and insert clearer transitional sentences between each. For instance, restructuring a sentence like "It is reported that cognitive functions have improved after training based on the combination of physical and cognitive exercises compared to any of them separately" to "Recent reports indicate that combining physical and cognitive exercises enhances cognitive functions more effectively than either approach alone" can strengthen the connection with the following sentence.
2. Consistent terminology is necessary when explaining the same concept. When using both "neurotrophic methods" and "neurotrophins," it's important to select one term and maintain consistency throughout. For example, using "neurotrophic factors" instead of "neurotrophic methods" can help maintain overall terminological consistency.
3. Strengthening logical connections between sentences will help readers better understand the content. By rephrasing a sentence like "Neurophysiological changes associated with rehabilitation are a relatively new area of research" to "Understanding neurophysiological changes associated with rehabilitation remains a burgeoning area of research," the relevance to the preceding sentence can be emphasized.
[METHOD]
1. The criteria for patient selection and exclusion in the clinical trial are not clearly defined. Particularly, the explanation regarding patient selection criteria is complex and may be difficult for readers to comprehend. The patient selection criteria and exclusion criteria should be summarized in a concise and clear manner to facilitate understanding for the readers. For instance, summarizing the patient selection criteria in a simple list format and providing clearer explanations for the exclusion criteria would be beneficial.
2. The description of EEG data collection and analysis methods is insufficient. Specifically, there is a lack of detailed information on the data processing and analysis procedures, which may hinder understanding of the study's reliability and reproducibility. Detailed explanation of EEG data collection and analysis methods is necessary to allow for the replication of the study process. Providing information on the software and algorithms used for data processing and analysis, as well as considering additional experiments or statistical methods to validate the analysis results, would be advantageous.
3. Insufficient information is provided regarding the measurement methods and accuracy of serum factors. Additional information is needed to demonstrate the reliability and accuracy of the measurement equipment and methods used in the experiment. Detailed explanation of the measurement methods and equipment used for serum factors is essential to enhance the reliability of the experimental results. Describing quality control and verification procedures for the measurement equipment, along with conducting additional experiments or comparative studies to demonstrate the consistency and accuracy of the results, is necessary.
[DISCUSSION]
1. The interpretation of results and discussions presented in the manuscript lacks sufficient clarity. For instance, there is a need for a clearer explanation regarding the potential slight reduction in the severity of brain damage in the patient group subjected to MTT. This requires more detailed explanation along with the interpretation of the significance and clinical implications of the results.
2. The relevance of the results presented in the manuscript to existing literature needs to be emphasized further. For example, a clearer explanation is needed to establish the association between the decrease in BDNF levels and cognitive impairment after cardiac surgery, along with discussing the consistency with previous studies.
3. More specific mention is required regarding future research directions based on the results of the manuscript. Particularly, there is a need to emphasize the necessity for further investigation into the mechanisms of brain damage during cardiac surgery and the impact of MTT on reducing brain damage. This would provide new insights into the field and enhance knowledge of clinical practice.
Author Response
[INTRODUCTION]
- To enhance the manuscript's flow to be more consistent and natural, there is a need to diversify similar sentence structures and insert clearer transitional sentences between each. For instance, restructuring a sentence like "It is reported that cognitive functions have improved after training based on the combination of physical and cognitive exercises compared to any of them separately" to "Recent reports indicate that combining physical and cognitive exercises enhances cognitive functions more effectively than either approach alone" can strengthen the connection with the following sentence.
RESPONSE: The sentence was replaced as recommended by the honorable reviewer.
- Consistent terminology is necessary when explaining the same concept. When using both "neurotrophic methods" and "neurotrophins," it's important to select one term and maintain consistency throughout. For example, using "neurotrophic factors" instead of "neurotrophic methods" can help maintain overall terminological consistency.
RESPONSE: We made changes to the terminological consistency.
- Strengthening logical connections between sentences will help readers better understand the content. By rephrasing a sentence like "Neurophysiological changes associated with rehabilitation are a relatively new area of research" to "Understanding neurophysiological changes associated with rehabilitation remains a burgeoning area of research," the relevance to the preceding sentence can be emphasized.
RESPONSE: The sentence was replaced as recommended by the honorable reviewer.
[METHOD]
- The criteria for patient selection and exclusion in the clinical trial are not clearly defined. Particularly, the explanation regarding patient selection criteria is complex and may be difficult for readers to comprehend. The patient selection criteria and exclusion criteria should be summarized in a concise and clear manner to facilitate understanding for the readers. For instance, summarizing the patient selection criteria in a simple list format and providing clearer explanations for the exclusion criteria would be beneficial.
RESPONSE: We corrected the Methods section. We would also like to note that the study sample was divided into two groups comparable in terms of clinical characteristics.
- The description of EEG data collection and analysis methods is insufficient. Specifically, there is a lack of detailed information on the data processing and analysis procedures, which may hinder understanding of the study's reliability and reproducibility. Detailed explanation of EEG data collection and analysis methods is necessary to allow for the replication of the study process. Providing information on the software and algorithms used for data processing and analysis, as well as considering additional experiments or statistical methods to validate the analysis results, would be advantageous.
RESPONSE: We provided the description of EEG data collection and analysis methods. We recorded EEGs via a 62‐channel Quik‐cap using NEUVO‐64 system (Compumedics, El Paso, TX, USA) at 2–3 days before surgery and at 11-12 days after CABG. The EEGs were recorded in the eyes-closed condition, in a dimly lit, soundproof, electrically shielded room, and recording lengths were about 5 min. The modified 10/10 system was used to locate the electrodes on the scalp. The reference and ground electrodes were placed on the tip of the nose and the center of the forehead respectively. To monitor eye movement artifacts, bipolar eye movement electrodes were placed on the canthus and cheek bone. The amplifier bandwidths were 1.0 to 50.0 Hz, and EEGs were digitized at 1000 Hz. The Neuroscan 4.5 software program (Compumedics, El Paso, TX, USA) was used to analyze the data offline using automatic algorithms. Any eye movements, electromyographic and other artifacts were visually inspected. Artifact-free EEG fragments were divided into 2 s epochs and underwent Fou-rier transformations. For each subject, the EEG power values were averaged within the theta range (4–6 Hz). The EEG frequency band selected for the planned analysis was de-termined by previous studies and the significance of low-frequency activity in brain damage [19, 34]. The 56 leads data were averaged into 5 electrode zones symmetrically in the left and right hemispheres: frontal (Fp1/2, AF3/4, F1/2, Fp3/4, Fp5/6, F7/8), central (FC1/2, FC3/4, FC5/6, C1/2, C3/4, C5/6), parietal (CP1/2, CP3/4, CP5/6, P1/2, P3/4, P5/6, P7/8), occipital (PO3/4, PO5/6, PO7/8, O1/2) and temporal (FT7/8, T7/8, TP7/8). The midline sites (Fpz, Fz, etc.) were excluded [34]. The indicator of relative change (Δ) was calculated using the formula: (baseline value–postoperative value)/baseline value) × 100%.
- Insufficient information is provided regarding the measurement methods and accuracy of serum factors. Additional information is needed to demonstrate the reliability and accuracy of the measurement equipment and methods used in the experiment. Detailed explanation of the measurement methods and equipment used for serum factors is essential to enhance the reliability of the experimental results. Describing quality control and verification procedures for the measurement equipment, along with conducting additional experiments or comparative studies to demonstrate the consistency and accuracy of the results, is necessary.
RESPONSE: We corrected the information about the measurement methods and accuracy of neurotrophic factors.
[DISCUSSION]
- The interpretation of results and discussions presented in the manuscript lacks sufficient clarity. For instance, there is a need for a clearer explanation regarding the potential slight reduction in the severity of brain damage in the patient group subjected to MTT. This requires more detailed explanation along with the interpretation of the significance and clinical implications of the results.
- The relevance of the results presented in the manuscript to existing literature needs to be emphasized further. For example, a clearer explanation is needed to establish the association between the decrease in BDNF levels and cognitive impairment after cardiac surgery, along with discussing the consistency with previous studies.
- More specific mention is required regarding future research directions based on the results of the manuscript. Particularly, there is a need to emphasize the necessity for further investigation into the mechanisms of brain damage during cardiac surgery and the impact of MTT on reducing brain damage. This would provide new insights into the field and enhance knowledge of clinical practice.
RESPONSE: We thank so much to the honorable Reviewer for his comments. We were able to improve our manuscript, based on these criticisms. We expanded and corrected the Discussion section.

Reviewer 2 Report
Comments and Suggestions for Authors
First of all, I would like to thank the authors for their innovative approach to a complex and recently more frequent condition: cognitive decline after cardiac surgery and understanding their complex mechanisms. But unfortunately I have some serious concerns:
1) MoCA could not be used to address if there is a clinical response in relation to any cognitive intervention. In those cognitively unimpaired its sensibility to detect minor cognitive changes is too small, and a comprenhensive neuropshychological assesment before and after the surgery should have been performed. In turn, it is necessary to considerate the possible "learning effect" that can occur and to adaptate the evaluation.
2) The sample is very heterogeneous in age and probabily in copathologies. Do we have information about neuroimaging features? There could be differences between those with cerebrovascular pathology, preclinical or prodromal stages of neurodegenerative disorders not properly assessed by MoCA test? Differences among drugs used? You have to consider that EEG can easily be modified in relation to many factors including hours of sleep, type of drugs and quantity of them...
3) EEG is a powerful tool in clinical practice, but in introduction it is suggested that could be as useful as those core biomarkers of AD or other cognitive disorders and that is not obviously true. EEG is good to assess for status epileptus, interictal epileptiform disacharges... but the modifications detected in relation to cognitive status are heteregenous, very variable in relation to many environmentla conditions... and this must be corrected.
4) Why not use synaptic integrity biomarker measurement (for instance NPTX2, because EEG performance should be more related to it)? If you are checking neurovascular integrity why not check differences in diffusion sequence in MRI (more available in clinical practice)? Why not use NfL, the most well known biomarker of axonal damage?
Why not perform a more long longitudinal follow-up to see if the differences are mantained?
Author Response
We thank to the honorable Reviewer for giving us the opportunity to submit a revised draft of our manuscript. We appreciate the time and effort that you have dedicated to providing your valuable feedback on our manuscript. We thank to the honorable Reviewer for the detailed review of our article.
- MoCA could not be used to address if there is a clinical response in relation to any cognitive intervention. In those cognitively unimpaired its sensibility to detect minor cognitive changes is too small, and a comprenhensive neuropshychological assesment before and after the surgery should have been performed. In turn, it is necessary to considerate the possible "learning effect" that can occur and to adaptate the evaluation.
Response: We agree with honorable reviewer that MoCA scale should not be used for this purpose. We used a cognitive screening with MoCA to exclude patients with severe cognitive impairment, the assessment was conducted upon admission to the hospital. The subjects with MoCA score ≥18 were excluded from the study. This is mentioned in the section 2.2. Neurophysiological assessment. We would also like to note that our study aimed to investigate the relationship between intervention-associated changes in markers of the neurovascular unit and EEG activity in cardiac surgery patients with and without multi-tasking training. This study did not include the evaluation of the impact of multitasking training on cognitive functioning. These results we published earlier (Tarasova I, Trubnikova O, Kukhareva I, Syrova I, Sosnina A, Kupriyanova D, Barbarash O. A Comparison of Two Multi-Tasking Approaches to Cognitive Training in Cardiac Surgery Patients. Biomedicines. 2023 Oct 18;11(10):2823. doi: 10.3390/biomedicines11102823).
- The sample is very heterogeneous in age and probabily in copathologies. Do we have information about neuroimaging features? There could be differences between those with cerebrovascular pathology, preclinical or prodromal stages of neurodegenerative disorders not properly assessed by MoCA test? Differences among drugs used? You have to consider that EEG can easily be modified in relation to many factors including hours of sleep, type of drugs and quantity of them...
Response: We thank the honorable reviewer for his detailed analysis of our manuscript. The MTT patients and controls were comparable in terms of age, baseline cognitive status, educational attainment, and other clinical factors. This information is provided in the Table 1.
The treatment protocol and medication did not have any significant differences.
1.5 T Exelart Atlas MRI system (Toshiba, Japan) was used to assess the baseline brain state in all patients. The leukoaraiosis, cysts, gliosis, and enlargement of cerebrospinal fluid spaces were examined. We added the information in the Methods section.
All pre- and post-intervention EEGs were collected at the same time of day and under the same conditions.
- EEG is a powerful tool in clinical practice, but in introduction it is suggested that could be as useful as those core biomarkers of AD or other cognitive disorders and that is not obviously true. EEG is good to assess for status epileptus, interictal epileptiform disacharges... but the modifications detected in relation to cognitive status are heteregenous, very variable in relation to many environmentla conditions... and this must be corrected.
Response: We agree with the honorable reviewer that EEG is a powerful tool in clinical practice. Increases in powers of slow oscillations in resting state EEG, especially theta oscillations, have been associated with cognitive deficits and cerebrospinal fluid total tau accumulation. It may possibly reflect the vulnerability or resilience of subcortical and thalamocortical loops and the ascending activating systems as additional and supplementary relevant information concerning the progression of cognitive decline. (Babiloni, C, Arakaki X, Azami H, Bennys K, Blinowska K, Bonanni L, Bujan A, Carrillo MC, Cichocki A, de Frutos-Lucas J, Del Percio C, Dubois B, Edelmayer R, Egan G, Epelbaum S, Escudero J, Evans A, Farina F, Fargo K, Fernández A, Ferri R, Frisoni G, Hampel H, Harrington MG, Jelic V, Jeong J, Jiang Y, Kaminski M, Kavcic V, Kilborn K, Kumar S, Lam A, Lim L, Lizio R, Lopez D, Lopez S, Lucey B, Maestú F, McGeown WJ, McKeith I, Moretti DV, Nobili F, Noce G, Olichney J, Onofrj M, Osorio R, Parra-Rodriguez M, Rajji T, Ritter P, Soricelli A, Stocchi F, Tarnanas I, Taylor JP, Teipel S, Tucci F, Valdes-Sosa M, Valdes-Sosa P, Weiergräber M, Yener G, Guntekin B. Measures of resting state EEG rhythms for clinical trials in Alzheimer's disease: Recommendations of an expert panel. Alzheimers Dement. 2021;17(9):1528-1553. https://doi.org/10.1002/alz.12311).
- Why not use synaptic integrity biomarker measurement (for instance NPTX2, because EEG performance should be more related to it)? If you are checking neurovascular integrity why not check differences in diffusion sequence in MRI (more available in clinical practice)? Why not use NfL, the most well known biomarker of axonal damage?
Response: Thank you very much for your comment. We have found one article dedicated to this problem, which is interesting (Huang XF, Xu MX, Chen YF, Lin YQ, Lin YX, Wang F. Serum neuronal pentraxin 2 is related to cognitive dysfunction and electroencephalogram slow wave/fast wave frequency ratio in epilepsy. World J Psychiatry. 2023 Oct 19;13(10):714-723. doi: 10.5498/wjp.v13.i10.714.). This possibility will be considered in the future.
Why not perform a more long longitudinal follow-up to see if the differences are mantained?
Response: We will plan to study longitudinal follow-up in the future to see if differences in EEG, cognitive functioning, and neurovascular unit markers are maintained.

Round 2
Reviewer 2 Report
Comments and Suggestions for Authors
I would like to thank the authors for try to answer most of my concerns. Obviously they are not able to modify sample selection and include those more properly phenotyped clinically (neurological and neuropsychological assesment) and biologically; and they didn not consider to address other potential physiopathogenic mechanisms involved and its potential biomarkers (such as those related with synaptic or axonal damage). I would only suggest to include in the text of the manuscript a section of future directions.